# Uterine Flushing Fluid-Derived Let-7b Targets CXCL10 to Regulate Uterine Receptivity in Goats during Embryo Implantation

**DOI:** 10.3390/ijms24032799

**Published:** 2023-02-01

**Authors:** Xinnuan Ning, Jie Li, Hui Fang, Siyuan Yu, Hongxia Zhang, Yanan Zhao, Lu Zhang, Aihua Wang, Yaping Jin, Dong Zhou

**Affiliations:** 1College of Veterinary Medicine, Northwest A&F University, Xianyang 712100, China; 2Key Laboratory of Animal Biotechnology, Ministry of Agriculture and Rural Affairs, Northwest A&F University, Xianyang 712100, China

**Keywords:** uterine flushing fluid, exosome, embryo implantation, miRNA, chi-let-7b-5p, CXCL10

## Abstract

Exosomes have the ability to carry a wide range of chemicals, convey them to target cells or target regions, and act as “messengers.” For the purpose of investigating embryo attachment, it is helpful to comprehend the range of exosomal mRNAs and miRNAs derived from the uterine flushing fluid before and after embryo attachment. In this study, we recovered exosomes from goat uterine rinsing fluid at 5, 15, and 18 days of gestation and used RNA-Seq to identify the mRNA and miRNA profiles of exosomes obtained from uterine rinsing fluid before and after embryo implantation. In total, 91 differently expressed miRNAs and 27,487 differentially expressed mRNAs were found. The target genes predicted by the differentially expressed miRNAs and the differentially expressed mRNAs were mainly membrane-related organelles with catalytic activity, binding activity, transcriptional regulation activity, and involved in metabolism, biological regulation, development, and other processes. This was revealed by GO analysis. Furthermore, KEGG analysis revealed that they were abundant in signaling pathways associated with embryo implantation, including the “PI3K-Akt signaling pathway,” “Toll-like receptor signaling pathway,” “TGF-beta signaling route,” “Notch signaling pathway,” and others. Moreover, our research has demonstrated, for the first time, that chi-let-7b-5p specifically targets the 3’UTR of CXCL10. Our research offers a fresh viewpoint on the mechanics of embryo attachment.

## 1. Introduction

An important step in the beginning of mammalian pregnancy is embryo implantation. It describes the period of time when the fertilized egg transforms into a blastocyst in the uterus, forms a temporary connection with the mother’s uterus, and is stationed there. The embryo and the mother’s endometrium engage in sophisticated communication during the engraftment stage [1]. According to previous studies, 75% of early mammalian embryo losses take place both before and after implantation [2,3]. Therefore, the key to a successful mammalian pregnancy is research of the embryo implantation stage.

Exosomes are small vesicles with a characteristic lipid bilayer structure and a diameter ranging from 30 to 100 nm. Exosomes were given their current name in 1987 after being discovered for the first time in sheep reticulum cells in 1983. Endocytosis, which collects DNA, RNA, protein, and other molecules in the cytoplasm into vesicular endosomes, produces exosomes, which are eventually released from the cell when they fuse with the plasma membrane [4,5]. Exosomes can convey the substances they contain by coming into contact with the cell membrane or by interacting with particular cell surface receptors. Exosomes are vital for the movement of materials, information transfer, immunological control, inflammatory response, and the onset and progression of cancer, according to previous studies [6,7]. For example, in pregnancy-related diseases, the differential substances released by the exocrine body can be used as a marker for the preliminary diagnosis of the disease [8]. In addition, exosomes can also be used as a drug carrier [9]. Blood, urine, amniotic fluid, and other body fluids contain a large number of exocrine bodies [10,11]. In 2013, Ng et al. extracted the uterine exocrine body from human uterine fluid for the first time [12]. Later, some researchers found evidence of exosomes in various animal species [13].

Numerous studies have demonstrated that exosomes can encourage intercellular communication in the female reproductive tract [14], and the proteins and miRNAs that exosomes carry are directly associated to activities such as embryo adhesion, invasion, and endometrial receptivity, among others [15]. Small non-coding RNAs called microRNAs (miRNAs), which are 20–25 nt in length and do not encode proteins, belong to this class. They are widely distributed in both plants and animals, mostly in methods that interact fully or partially with the target mRNA. During transcriptional or post-transcriptional processes, the 3′UTR, 5′UTR, or open reading frame negatively regulates gene expression by deteriorating the target mRNA or preventing its translation. Nearly every cell activity process involves miRNAs. They are crucial for cell differentiation, animal development, and homeostasis maintenance. In addition, several miRNAs can serve as possible cancer therapeutic targets and prognostic markers [16]. There have been very few studies in recent years on the miRNA and mRNA profiles of exosomes in uterine rinsing fluid before and after goat embryo implantation, despite an increase in studies on the expression variations of proteins, miRNAs, and mRNAs transported by exosomes under various situations. We used RNA sequencing (RNA-Seq) technology to compare the miRNA and mRNA expression alterations of exosomes in uterine rinse fluid before and after embryo implantation. Reihart et al. [17] initially identified the let-7 family as a relatively conservative miRNA. Several studies have found that members of the let-7 family are crucial for mammalian embryonic development [18] and embryo implantation [19]. In our study, let-7b-5p expression reduced noticeably as pregnant days increased, and let-7 was abnormally expressed in numerous disorders. As an example, let-7c and let-7g expression were raised in patients with head and neck squamous cell carcinoma [20] and bladder cancer [21], while some studies discovered that the expression of let-7 family genes was reduced in patients with uterine leiomyoma [22]. Recent research has revealed that members of the let-7 family are crucial for the growth [18] and implantation [19] of mammalian embryos; however, there is still uncertainty regarding their function in the implantation of goat embryos. We investigated how let-7b-5p affected the growth of goat endometrial epithelial cells and the expression of genes involved in goat embryo implantation in this study. In addition, we established, for the first time, that let-7b-5p directly targets CXCL10. Our research offers a fresh perspective on the embryo implantation research mechanism.

## 2. Results

### 2.1. Exosome Identification

On the fifth, fifteenth, and eighteenth days of pregnancy, exosomes were recovered from goat uterine flushing fluid by ultracentrifugation. The morphology, diameter distribution, and concentration distribution of exosomes were studied using a transmission electron microscope and a nanoparticle size tracking analyzer. In the transmission electron microscope image of the exosomes, a characteristic lipid bilayer membrane structure was visible (Figure 1A). On the fifth day of pregnancy, the exosomes isolated from the goat uterine flushing fluid had an average particle size of 75.81 nm and a concentration of 1.81 × 10^10^ particles/mL; on the fifteenth, they were 77.29 nm and 1.22 × 10^11^ particles/mL; and on the eighteenth, they were 75.24 nm and 8.96 × 10^10^ particles/mL (Figure 1B,C). Exosome-specific protein markers (TSG101 and CD63) were detected by Western blotting and were shown to be positive (Figure 1D). Based on these findings, we were able to separate exosomes from the goat uterine flushing fluid.

### 2.2. Differential mRNA and miRNA Expression Analysis

Using high-throughput sequencing techniques, we evaluated the variations of exosome mRNAs and miRNAs in uterine flushing fluid at 5, 15, and 18 days of gestation. In this study, 27,487 differentially expressed mRNAs and 91 differentially expressed miRNAs were found. Of these, 7352 differentially expressed mRNAs were upregulated in the P_D15 and P_D18 groups, while 1868 were downregulated. Likewise, 20 differentially expressed miRNAs were upregulated, while 10 were downregulated. Moreover, 13 differently expressed miRNAs were upregulated, and 13 were downregulated in the P_D5 and P_D15 groups, respectively, as were 1654 upregulated mRNAs and 8981 downregulated mRNAs. There were 14 and 4718 downregulated miRNAs and mRNAs, respectively, and 21 and 2644 upregulated miRNAs in the P_D5 and P_D18 groups (Figure 2A). The differentially expressed genes in the two groups were also compared. In the P_D15 and P_D18 groups, there were 2 and 493 differently expressed miRNAs and mRNAs, respectively, while there were 3 and 834 in the P_D5 and P_D18 groups and 2 and 414 in the P_D5 and P_D15 groups. The three groups shared 4 and 2974 distinct mRNAs and miRNAs, respectively (Figure 2B). The Venn diagram revealed that there were 4 miRNAs and 493 mRNAs that were specifically differentially expressed in the P_D5 and P_In D18 group, 414 mRNAs and 2 miRNAs in the P_D5 and P_In D15 group, and 834 mRNAs and 2 miRNAs in the P_D15 and P_In D18 group. There were 2947 mRNAs and 4 miRNAs that were differentially expressed across the three groups (Figure 2C,D). On these differentially expressed mRNAs and miRNAs, cluster analysis was performed. The expression of miRNAs and mRNAs (the mRNAs with a 14-fold variation in expression were chosen for cluster analysis) in exosomes isolated from goat maternal uterine fluid at various gestational days were different, as shown by the heat map (Figure 3).

### 2.3. GO Enrichment and KEGG Pathway Analysis of Differentially Expressed Genes

In different gestational days of each group, we annotated the target genes of differentially expressed mRNAs and miRNAs into the biological processes, cellular components, and molecular functions of the GO database. The primary functions of the target genes of these differentially expressed mRNAs and miRNAs were “metabolic process,” “biological regulation,” “cellular process,” and “cellular component of biological processes,” as well as “organization or biogenesis,” “developmental process,” “multi-organism process,” “response to stimulus,” “localization,” and “immune system process,” among others. Cell junctions, cells, membrane-enclosed lumens, membranes, cell parts, organelle parts, protein-containing complexes, etc. were the key cell components involved. Molecular transducer activity, catalytic activity, binding, antioxidant activity, structural molecule activity, transporter activity, molecular function regulator activity, etc. were the main molecular functions (Figure 4 and Figure 5).

Using the KEGG database, signal pathway annotation of differentially expressed mRNA and target genes of differentially expressed miRNA was carried out in each group (only the top 20 are shown). The main mRNAs that differed in expression were those related to “Metabolic pathways,” “Oxidative phosphorylation,” “Spliceosome,” “Thermogenesis,” “Parkinson disease,” etc.; “Metabolic pathways,” “Huntington disease,” “Carbon metabolism,” “Spliceosome,” “Thermogenesis,” etc.; and “Pathways in cancer,” “Phosphatidylinositol signaling system. “The “FoxO signaling pathway,” the “Apelin signaling pathway,” “Adherens junction,“ “Thyroid hormone signaling system,“ “Pathways in cancer, “ etc. were the key signaling pathways where the target genes of miRNAs that changed between days 15 and 18 of gestation were expressed. The “AGE-RAGE signaling route in diabetic problems,“ “TGF-beta signaling pathway,“ “Axon guidance signaling pathway,“ “Rap1 signaling pathway, “ “Platelet activation, “ “PI3K-Akt signaling pathway,“ and “Toll-like receptor signaling pathway“ were the primary complications between days 5 and 15 of pregnancy. Target genes of differential miRNAs were primarily involved in “Axon guidance,” “Protein digestion and absorption,” “Legionellosis,” “Transcriptional misregulation in cancer,” “Leukocyte transendothelial migration,” “Biosynthesis of unsaturated fatty acids,” “Adherens junction,” “AMPK signaling pathway,” etc. between 5 and 18 days of gestation (Figure 6). These findings imply that the miRNA and mRNA that are variably expressed in exosomes have distinct regulatory mechanisms at various gestational stages.

### 2.4. Validation of Deep Sequencing Results by Quantitative RT-PCR

The miRNAs (chi-miR-199b-5p, chi-miR-136-5p, and chi-miR-106b-3p) and mRNAs (*FTH1*, *PTGS2*, *CTNNB1*, *ACSL4*, and *GPX4*) that were shown to be differentially expressed above were verified using RT-PCR. The results matched the sequencing information (Figure 7).

### 2.5. Expression and Localization of chi-let-7b-5p

Members of the let-7 family are crucial for the growth and implantation of mammalian embryos [18,19], and our sequencing findings reflect changes in their expression. On days 5, 15, and 18 of pregnancy, only chi-let-7b-5p showed a tendency of differential expression (Appendix A). Therefore, we investigated the potential effects of chi-let-7b-5p on goat embryo implantation. In order to further verify the expression level of miRNA and determine the location of chi-let-7b-5p in goat endometrium, we detected it by RT-PCR and in situ hybridization. In tissues, the expression of chi-let-7b-5p decreased with the increase of pregnancy days (Figure 8A). The results of in situ hybridization showed that chi-let-7b-5p was highly expressed in endometrial epithelial cells, luminal epithelial cells, and stromal cells in tissues on the 5th day of pregnancy, and the expression was significantly decreased on the 15th day of pregnancy, while chi-let-7b-5p was not detected in endometrial cells or stromal cells on the 18th day of pregnancy. According to the results of in situ hybridization, chi-let-7b-5p was highly expressed in tissues on the fifth day of pregnancy in endometrial epithelial cells, luminal epithelial cells, and stromal cells. On the 15th day of pregnancy, the expression significantly decreased, and on the 18th day of pregnancy, chi-let-7b-5p was not found in endometrial cells or stromal cells (Figure 8B).

### 2.6. Effects of chi-let-7b-5p on Proliferation of gEEC and Expression of Genes Related to Embryo Implantation

Goat endometrial epithelial cells were transfected with chi-let-7b-5p mimic or inhibitor, and the transfection efficiency was greater at concentrations of 100 nM and 150 nM, respectively (Figure 9A). These concentrations were used in follow-up studies. According to CCK8 data, neither the chi-let-7b-5p mic group nor the inhibitor group had any impact on the proliferation of goat endometrial cells compared to the NC group (Figure 9B). We employed RT-PCR to measure the expression of genes associated with embryo implantation in order to confirm the impact of chi-let-7b-5p on goat embryo implantation. As seen in Figure 9C, inhibition of chi-let-7b-5p considerably enhanced the expression of *CXCL10* and *MUC1*, whereas the chi-let-7b-5p mimic dramatically lowered their expression (*p* < 0.01). *MUC1* has been described as a target gene for members of the let-7 family [23], while *CXCL10* has been described as a crucial cytokine for the implantation of goat embryos. These findings suggest that chi-let-7b-5p inhibits *CXCL10*, which has an impact on the implantation of goat embryos.

### 2.7. Expression and Localization of CXCL10

Through the miRNA target gene online prediction software, we found that there was a binding site between chi-let-7b-5p and *CXCL10* (Figure 10A). We detected its expression in goat uterine tissues by RT-PCR and immunohistochemistry. Compared with the tissues on the 5th and 15th days of pregnancy, *CXCL10* mRNA level in tissues on the 18th day of pregnancy was significantly upregulated (*p* < 0.01) (Figure 10B), and the expression pattern was opposite to chi-let-7b-5p. Immunohistochemical results showed that it was mainly located in endometrial luminal epithelial cells, glandular epithelial cells, stromal cells, and deep muscle layers (Figure 10C).

### 2.8. CXCL10 Is the Direct Target Gene of chi-let-7b-5p

We performed the twofold luciferase reporting experiment to confirm that chi-let-7b-5p and CXCL10 have a targeting interaction. A 430 bp fragment with the CXCL10 3’-UTR binding site was amplified and cloned into the psiCHECK2 vector where it was given the name psiCHECK2-CXCL10 WT. The mutant sequence of CXCL10 3’UTR was obtained by fusion PCR, and the created mutation vector was given the name psiCHECK2-CXCL10 MUT (Figure 11A). Sequencing results confirmed that the sequence was correct (Figure 11B). When the chi-let-7b-5p mimic or NC mimic was co-transfected with psiCHECK2, psiCHECK2-CXCL10 WT, or psiCHECK2-CXCL10 MUT into HEK293T cells, we discovered that the former reduced luciferase activity of the latter by 63% (Figure 11C), demonstrating the connection between chi-let-7b-5p and CXCL10.

In addition, the ELISA results demonstrated that the chi-let-7b-5p mimic decreased the expression of the CXCL10 protein level (*p* < 0.01), but the chi-let-7b-5p inhibitor would have resulted in the upregulation of CXCL10 expression in gEECs (*p* < 0.01, Figure 11D).

## 3. Discussion

Numerous cells can create exosomes both normally and in malignant situations. Exosomes serve as crucial “messengers” in the development of mammalian embryos and carry a significant amount of biological information materials. Exosomes can be categorized into three groups in the study of embryonic development: exosomes derived from the embryo, placenta, and the mother. Saadeldin et al. [24], in their research on in vitro embryo culture, revealed, for the first time, that in vitro embryos could take up exosomes from nearby embryos, which included mRNAs including *Oct4*, *Sox2*, *cMyc*, and others that might encourage embryo development. Fasl and TRAIL-containing exosomes from the placenta may have a role in immune control and embryo protection [25]. Exosomes from the mother and the umbilical cord have been reported to encourage endothelial cell migration and proliferation [26], which is crucial for sustaining a healthy pregnancy while the embryo develops. In a previous study, exosomes were produced using the hormone treatment of human endometrial cancer cells. The authors discovered that exosomes dramatically improved cell adherence and proliferation as well as embryonic development and implantation [27]. Exosomes obtained under various circumstances have had their profiles of miRNAs, cirRNAs, mRNAs, and lncRNAs examined in recent years. Researchers discovered 272 miRNAs that were differently expressed in exosomes from endometritis-affected cattle [28]. Exosomes were extracted from blood samples of Holstein cattle for 30 days of pregnancy by Markkandan et al. They discovered 29 differently enriched miRNAs in the gestation group compared to the normal group [29]. Exosomes from endometriosis have been found to contain 938 lncRNAs, 39 miRNAs, and 1449 mRNAs with overlapping differential expression [30]. Exosomes contain concentrated and persistent circulating RNAs that can be found in urine [31]. In exosomes extracted from the serum of alcohol-dependent patients, 254 circRNAs were recently discovered to be differentially expressed, of which 149 circRNAs were upregulated and 105 were downregulated. These circRNAs may control neuronal projection and axonal regeneration, according to GO analysis [32]. Studies on the miRNA and mRNA spectrum of exosomes under various goat pregnancy circumstances, however, are still flawed.

Exosomes were extracted from goat uterine lavage on the fifth, fifteenth, and eighteenth days of pregnancy for this investigation. We used high-throughput sequencing technology to examine the differential changes in exosome mRNAs and miRNAs in goat uterine lavage fluid on different days of pregnancy after detection by the exosome identification method. This analysis gave us a new hypothesis for the study of the mechanism of goat embryo implantation. Our sequencing findings revealed that the exosomes isolated from the samples at 5 and 15 days of gestation contained 26 differentially expressed miRNAs and 10,635 differentially expressed mRNAs, while the exosomes isolated from the samples at 5 and 18 days of gestation contained 35 differentially expressed miRNAs and 7362 differentially expressed mRNAs, respectively. The recovered exosomes from 15 and 18 days of gestation contained 9220 differently expressed mRNAs and 30 differentially expressed miRNAs, respectively. To determine which biological processes and signal pathways the target genes with differentially expressed miRNAs and mRNAs may be involved in, GO and KEGG were utilized. The GO analysis revealed that the predicted target genes of these differentially expressed miRNAs were membrane-related organelles with catalytic activity, binding activity, transcriptional regulation activity, and engaged in processes such as development, metabolism, and biological regulation. In addition, we used the KEGG database to annotate them, and certain pathways have been suggested to be involved in embryo attachment. The “Toll-like receptor signaling pathway,” “PI3K-Akt signaling pathway,” “TGF-Beta signaling pathway,” and “Notch signaling pathway” are a few examples. TGF- (Transforming Growth Factor) is recognized to participate in a variety of biological activities. In a previous study, the number of implanted embryos dropped on day 3 of gestation when mice were given a particular inhibitor of TGF-, the most specific signal transduction factor Smads, indicated that TGF- may be involved in the process of embryo implantation [33]. The pre-implantation stage [34], embryo survival, and the mouse embryo’s ability to utilize glucose [35] depended on the PI3K-Akt signaling pathway being activated. The crucial function of the PI3K-Akt signaling pathway in the mouse implantation window stage was clarified by Liu L et al. [36]. Toll-like receptor 4 (TLR) is spatiotemporally expressed on distinct days of early pregnancy in the uterus of sheep, suggesting its significance in sheep reproduction, as demonstrated in earlier investigations on human and mouse implantation. Endometrial physiological processes such endometrial decidualization, embryo implantation, and placental differentiation and development are all influenced by the Notch signaling pathway [37]. Notch1 levels are said to be essential for successful implantation and endometrial decidualization and are directly associated with endometrial receptivity and maternal-fetal interactions. Furthermore, Notch1 expression levels were dramatically elevated in normal women throughout the endometrial “receptivity” phase [38].

As the number of days of goat pregnancy increased in our study, the expression of chi-let-7b-5p decreased. Members of the let-7 family have been shown to express differently throughout the implantation stage in rats, and they were especially found in the glandular epithelium, luminal epithelium, and decidual cells [19]. Using in situ hybridization, it was discovered that chi-let-7b-5p was primarily expressed in the muscle layers, glandular epithelial cells, and luminal epithelial cells of goat uterine tissues. These cells are crucial to the process of embryo implantation and may also play a role in controlling embryo implantation. Later, we examined the impact of chi-let-7b-5p on the expression of genes involved in goat embryo implantation. *CXCL10* and *MUC1* expression was greatly reduced by the chi-let-7b-5p mimic, but it was significantly boosted by the chi-let-7b-5p inhibitor. One of the let-7 family’s target genes is *MUC1* [39], and *CXCL10* is a crucial cytokine for embryo implantation. We discovered a binding site between chi-let-7b-5p and *CXCL10* using online target gene prediction software for miRNAs. Furthermore, examination of the double luciferase reporter gene demonstrated that chi-let-7b-5p directly targeted *CXCL10*. During the implantation stage of a human embryo, the mother and the embryo will secrete chemokines that will act on trophoblast cells and encourage their migration [40]. In addition, CXCL10 can draw immune cells to the endometrium [41]. Numerous studies have demonstrated that let-7 miRNA [42,43] can be produced by animal embryos at the pre-implantation stage and that the lin28b/let-7 axis can regulate the differentiation of fetal Treg cells [44]. In addition, it has been demonstrated that the immune microenvironment during pregnancy is essential to the embryonic implantation process [45]. Let-7b-5p may come from embryos, and our research has demonstrated the direct targeted link between it and CXCL10. It also plays a critical function in embryo implantation by controlling the uterine immune milieu. More investigation is required to establish let-7b-5p’s function in the implantation of goat embryos.

## 4. Materials and Methods

### 4.1. Sample Collection and Exosome Isolation

Six sexually mature Guanzhong dairy goats were split into three groups at random, with two goats in each group (*n* = 2), and were kept in the Yangling Northwest Agriculture and Forestry University’s experimental animal center. The female goat and male goat freely mated after estrus, and this was noted as the 0th day of pregnancy. The pregnancy was determined by looking at the tubular or linear pregnancy in the uterus on the fifth day (P_D5, in the pre-implantation period), the fifteenth day (P_D15, in the pregnancy recognition period), and the eighteenth day (P_D18, in the embryo adhesion period). A midline laparotomy and hysterectomy were performed on the pregnant sheep on the fifth, fifteenth, and eighteenth days. The ewe’s uterus was then quickly collected (care was taken to guarantee minimal injury), and the cervix and horn were bound with ropes to prevent the fluids from pouring out. The ewe’s uterus was placed in a thermos bottle and kept at a temperature of 15–20 °C before being transported in 3–4 h to the lab. To preheat, we placed the sterilized normal saline in a water bath set to 35 °C. Then, we added the necessary dosage of penicillin to the uterus before washing it. We washed the outside wall of the uterus three times with regular saline after eliminating the contaminants from outside the uterus. A segment of infusion tube was cut, inserted into the uterus from the cervix, and connected to a 20 mL syringe, sucked, and repeated eight times. The flushed solution was collected into sterilized centrifuge tube, the membrane was sealed, and the samples were stored at 4 °C for standby.

Ultracentrifugation [46], density gradient centrifugation [47], immunomagnetic beads [48], ultrafiltration centrifugation [49], gel chromatography [50], PEG-base precipitation [23], and kit extraction are some of the techniques used to separate exocrine [46]. The technique that is most frequently used to separate exosomes is ultracentrifugation. Exosomes are separated using this method by more than 56% of exosome researchers, according to investigation and research [48]. Using ultracentrifugation, the exocrine body in the uterine flushing fluid was removed in this experiment. The collected uterine flushing solution was centrifuged at 4 °C for 30 min at 2000× *g* to remove macromolecular pollutants, dead cells, cell fragments, and other contaminants. A new, dry, sterile centrifuge tube was used to transfer the supernatant. Care was taken not to suction out the sediment. Then, at 45 min of centrifuging 12,000× *g* at 4 °C, a dry, sterile 39 mL ultracentrifugation tube was filled with the liquid (Beckman, CA, USA). Next, the solution underwent 110,000× *g* ultracentrifugation at 4 °C for 2 h. If the liquid was insufficient, then sterile PBS (PH 7.4) was added, and the ultracentrifugation tube was closed. Then, we the ultracentrifuge tube off using sterilizing scissors, removed the supernatant, and left 2 mL of liquid suspended sediment. We filtered the heavy suspension liquid with a 0.22 μm filter (JIN TENG, Tianjin, China) before transferring it to a 6.0 mL ultracentrifuge tube (Beckman, CA, USA), where PBS was refilled and sealed. Afterward, the solution underwent 110,000× *g* ultracentrifugation for 70 min at 4 °C. We threw the supernatant, resuspended the sediment in 1 mL of sterile PBS, then moved the heavy suspension to a 6.0 mL ultracentrifuge tube, filled it with PBS, and closed it. The solution underwent 110,000× *g* ultracentrifugation for 70 min at 4 °C. The precipitate was subpackaged and stored at −80 °C after being resuspended in the appropriate volume of sterile PBS.

### 4.2. Transmission Electron Microscopy

First, samples were sent for detection after 5 μL of exosomes produced by overspeed centrifugation were removed and diluted to 10 μL. Then, 10 μL of the sample was poured onto the copper mesh, where it precipitated for 1 min. Filter paper was used to get rid of the float. On the copper mesh, 10 μL of phosphotungstic acid was poured and allowed to precipitate for 1 min. In addition, filter paper absorbed the float. The samples were at room temperature for a while, and imaging at 80 kV was performed using an electron microscope (HITACHI, Tokyo, Japan).

### 4.3. Nanoparticle Tracking Analysis

Exosomes obtained by overspeed centrifugation were extracted, diluted to a volume of 30 μL, and sent for detection in samples. Only after the instrument performance test had been qualified with the reference material could the exosome sample be loaded. To prevent a sample from clogging the injection needle, close attention was paid to the gradient dilution. Following testing of the material, the exosome content and particle size could be determined using the NanoFCM equipment (NanoFCM, Xiamen, China).

### 4.4. Western Blot Analysis

To extract the exocrine protein, we used the complete protein extraction kit (KeyGEN BioTECH, Nanjing, China) and the BCA protein content kit to assess the protein concentration (KeyGEN BioTECH, Nanjing, China). We transferred the sample to a polyvinylidene fluoride (PVDF) membrane (Roche, Basel, Switzerland), sealed it with 5% skimmed milk powder for one hour, and then used the first antibody (TSG101, 1:1000; CD63, 1:1000, Abcam, Cambridge, UK) for an overnight incubation at 4 degrees and the second antibody (ZHHC, Shanxi, China) for a one-hour incubation at 1:5000 room temperature. We placed the film in the imager (Bio-Rad, Hercules, CA, USA) and exposed it after incubating it with ECL luminous liquid (DiNing, Beijing, China). Then, we saved the image.

### 4.5. RNA Extraction and Library Construction

In this work, total RNA was extracted from the exosomes using the miRNeasySerum/Plasma Kit (Qiagen, Düsseldorf, Germany). The exocrine body was removed from the −80 °C refrigerator and placed in a water bath at 37 °C to melt. Then, the 500 μL of the QIAL Lysis Agent was added, mixed upside-down, and left to incubate for 5 min at room temperature. We combined 100 μL of chloroform and vigorously shake for 15 s. Then, we left the solution to incubate for 3 min at room temperature. The solution underwent 15 min of centrifuging 12,000× *g* at 4 °C. Then, we added 1.5 times the volume of anhydrous ethanol to the upper water phase in a fresh centrifuge tube and mixed. We filled the RNeasy MinElute centrifuge column with the liquid from the previous step, centrifuged at 8000× *g* at room temperature for 15 s, and then discarded the liquid. Next, we added 700 μL of RWT buffer, 8000× *g*, centrifuged for 15 s at room temperature, and then removed the liquid. The centrifuge column was placed in a fresh centrifuge tube, which was centrifuged at 12,000× *g* for 5 min. The liquid and centrifuge tube were discarded. RNA was eluted by centrifuging the centrifuge column for 1 min at 12,000× *g* at room temperature while adding 14 μL RNase-free water to a fresh centrifuge tube. BGI carried out the quality control, library creation, and total RNA in exosomes sequencing.

### 4.6. Sequencing Data Analysis

The reference genome and transcriptome were compared with the filtered clean reads of the raw data produced from sequencing, respectively. Appendix A display the statistics on the quality of filtered reads for the samples. HISAT software [51] and comparison (mRNA) were used. The reads obtained by sequencing were compared with the reference genome (smallRNA) using Bowtie2-2.0.3 software [52]. MiRNAs and other non-coding RNAs were anticipated based on the location of the sequencing data on the genome. The sequencing data was annotated with non-coding RNA, and reads were also checked with non-coding RNA and Rfam databases. For comparison information, see Appendix A.

### 4.7. Differentially Expressed miRNAs and mRNAs

RSEM [53] was used to determine the levels of gene and transcript expression. This study used the DEGseq approach described in Wang L et al. [54] to carry out differential gene detection. Through Hochberg, Y., Benjamin, Y., Storey, J., and Tibshirani, R., the two methods were used to convert *p*-values to Q values correctly. We defined genes with more than twice the differential multiple and Q value 0.001 as significantly differentially expressed genes and screened them in order to increase the accuracy of DEGs. Various small RNA analyses were performed. A special molecular tag called UMI was initially added to the cDNA during library preparation in order to measure the small RNA. According to Audio S. et al. method’s for sequencing-based differential gene detection [55], which was published in Genome Research, in our analysis, differential expression genes were those that had an FDR of less than 0.001 and multiple differences of at least two. We used the R software’s pheatmap function for hierarchical clustering analysis in accordance with the results of the difference detection.

### 4.8. GO and KEGG Enrichment Analysis

The analysis of exosome differentially expressed genes by GO and KEGG was performed. We used RNAhybrid [56], miRanda [57], and TargetScan [58] to predict the target genes of the differentially expressed miRNAs. GO and KEGG analysis were then used to enrich the putative target genes. In GO analysis, the number of genes in each term of the Gene Ontology database was computed after all candidate genes were mapped to each phrase in the database. The GO terms that were highly enriched in differentially expressed genes as compared to the other gene backgrounds of the species were then identified using hypergeometric testing. The program “Go:: termfinder” was used to evaluate the mRNA that was differentially expressed. The threshold was set at Qvalue (corrected Pvalue) = 0.05 after the estimated *p*-value was corrected by Bonferroni [59]. The primary biological functions of differentially expressed genes were identified using a GO function significance enrichment analysis. A significant public database pertaining to Pathway is KEGG Pathway [60]. Based on KEGG Pathway, the significant enrichment analysis of Pathway was conducted. To determine which pathways were considerably enriched in candidate genes compared to the background of the entire genome, the hypergeometric test was utilized. Pathways that were highly enriched in differentially expressed genes had a Qvalue of 0.05 or lower. The most crucial biochemical metabolic and signal transduction pathways connected to the candidate genes can be identified using Pathway’s considerable enrichment.

### 4.9. RT-PCR

Following the manufacturer’s instructions, the total RNA was extracted using the Trizol (Takara, Japan) reagent. Evo M-MLV reverse transcription Kit (AG Bio, Changsha, China) was used to reverse-transcribe common mRNA, while the miRNA 1st strand cDNA synthesis Kit was used to reverse-transcribe miRNA (vazyme, Nanjing, China and AG Bio, Changsha, China). For the mRNA and miRNA analyses, real-time quantitative RT-PCR was carried out using the SYBR Green method (vazyme, Nanjing, China), utilizing GAPDH and U6 as the internal references, respectively. Data were analyzed using 2^−ΔΔCt^ method. All reactions were performed in at least three independent experiments. Table 1 displays the sequence information for each primer.

### 4.10. Fluorescence In Situ Hybridization

Initially, chi-let-7b-5p was discovered via in situ hybridization in goat endometrium. The chi-let-7b-5p probe with cy3 label was created and manufactured by Servicebio. Simply put, following dewaxing, rehydrating, and repairing antigens on paraffin sections, protease K (20 μg/mL) (Servicebio, Wuhan, China) digestion occurred for 20 min, followed by PBS flushing. Then, we added 3% methanol-H_2_O_2_ dropwise, incubated for 15 min without light at room temperature to inhibit endogenous peroxidase, and then rinsed with PBS. This process was followed by 1 h of pre-hybridization at 37 °C. Using the probe’s hybridization buffer as the negative control, the hybridization solution containing the chi-let-7b-5p probe was hybridized overnight at 42 °C before being washed with the washing buffer. Then, we added the hybridization solution containing Imaging oligo (DIG) dropwise, incubated at 42 °C for 3 h, and then washed with washing buffer solution. For 30 min, normal rabbit serum was sealed at room temperature, and anti DIG HRP was added after removing the sealing fluid. We incubated the solution at 37 °C for 40 min, then washed sections in PBS four times for 5 min each using FITC Tyramide Reagent. After 5 min of dark, room-temperature reaction, the sections were rinsed with PBS; After washing the film, DAPI was incubated in the dark for 8 min before being sealed with an anti-fluorescence quenching agent. The pictures were captured with a microscope (NIKON Tokyo, Japan).

### 4.11. Cell Transfection and CCK8

The chi-let-7b-5p mimic and inhibitor, as well as the corresponding NC mimic and inhibitor, were bought from the Guangzhou Ruibo biological company. Immortalized gEECs were obtained as previously described [61] The cells were transfected once they had fused to 50%. In brief, chi-let-7b-5p mimic was diluted with serum-free medium Opti, mixed with transfection reagent (ThermoFisher Scientific, Waltham, MA, USA), incubated at room temperature for 20 min, added to the appropriate well drop by drop, and collected after 48 h for follow-up operation. Following transfection, 10 μL CCK8 (AbMole Bioscience Inc., Houston, TX, USA) was added to each well at the appropriate time in order to study the impact of chi-let-7b-5p on the proliferation of gEEC. Incubation was then continued for another two hours. The absorbance of gEEC at 450 nm was measured using a Thermo Scientific Microplate Reader (ThermoFisher Scientific, Waltham, MA, USA).

### 4.12. Immunohistochemistry

Using immunohistochemistry (IHC), we evaluated the expression of CXCL10 in goat uterus. Slices were taken out of the refrigerator at 4 °C and baked for 30 min at 60 °C. To remove the paraffin from the slices, they were submerged in xylene and a succession of alcohol solutions of varying concentrations. Slices were placed in PH6.0 citrate buffer, heated to a boil in a microwave, and then cooked for 15 min on low heat. The sample was then washed three times with PBS for five minutes each after cooling to room temperature. Histochemical staining was carried out using the ready-to-use immunohistochemistry hypersensitivity UltraSensitive^TM^ SP Kit (Maixin-Biotech, Fuzhou, China). The negative control section was switched out for PBS, and the primary antibody CXCL10 (Abbexa, Cambridge, UK) was incubated at a 1:300 dilution. Hematoxylin was counterstained for 25 s while the DAB (Maixin-Biotech, Fuzhou, China) color development time was monitored under a microscope. After being exposed to a variety of alcohol and xylene concentrations, the slices were sealed with neutral gum.

### 4.13. Dual-Luciferase Reporter Assay

The goat CXCL10 gene’s 3′UTR (untranslated region) was amplified by PCR and introduced into the psiCHECK2 vector. The partial sequences for the CXCL10 3′UTR WT and MUT were 5′-AGAGAGAAGCTACCTCT-3′ and 5′-AGAGAAGAGCATCAT-3, respectively. We employed 293T cell lines that had been frozen in our lab for later transfection. One day prior to transfection, cells under favorable growth conditions and in the logarithmic growth phase were transferred to 24-well plates, and HEK293T cells were co-transfected with NC-mimic or chi-let-7b-5p mimic. According to the kit’s instructions, dual-Luciferase^®^ Reporter Assay System (Promega, WI, USA) was used to detect the firefly luciferase and renilla luciferase activity 36 h after transfection. The ratio of renilla to firefly luciferase was used to present the findings.

### 4.14. Statistical Analysis

Data are presented as the mean ± standard error of the mean (SEM). The mapping of the data was performed using Graphpad Prism 5.0. All data were analyzed using SPSS software. To examine the statistical significance of the data differences, the non-parametric *t*-test and one-way ANOVA were utilized. Moreover, *p* < 0.05 indicates a statistically significant difference, and *p* < 0.01 indicates an incredibly significant difference. We employed two methods that correct *p*-values to Q-values for the sequencing data developed by Benjamini, Hochberg, Storey, and Tibshirani. Significantly differentially expressed genes (SDEGs) were classified and screened as genes with a Q-value ≤ 0.001 and a difference greater than two times the control.

## 5. Conclusions

In conclusion, we found some exosomes in the uterine rinse fluid both before and after embryo implantation. We provide fresh light on mother-fetal communication during goat embryo implantation by contrasting the differential expression of mRNAs and miRNAs in exosomes recovered from uterine rinsing fluid before and after embryo implantation. Investigating the precise functions of these differently expressed mRNAs and miRNAs during embryo attachment will be the main goal of our future work.

## Figures and Tables

**Figure 1 ijms-24-02799-f001:**
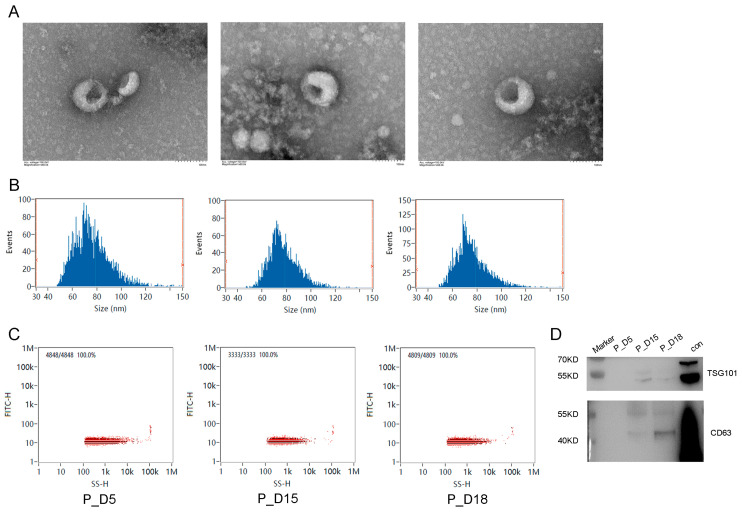
Exosome identification. (**A**) Electron micrograph. Scale bar, 100 nm. (**B**) Particle size distribution. (**C**) Exosomes concentration distribution. From left to right, the samples are P_D5, P_D15 and P_D18. (**D**) TSG101 and CD63 exosome markers were discovered using Western blotting.

**Figure 2 ijms-24-02799-f002:**
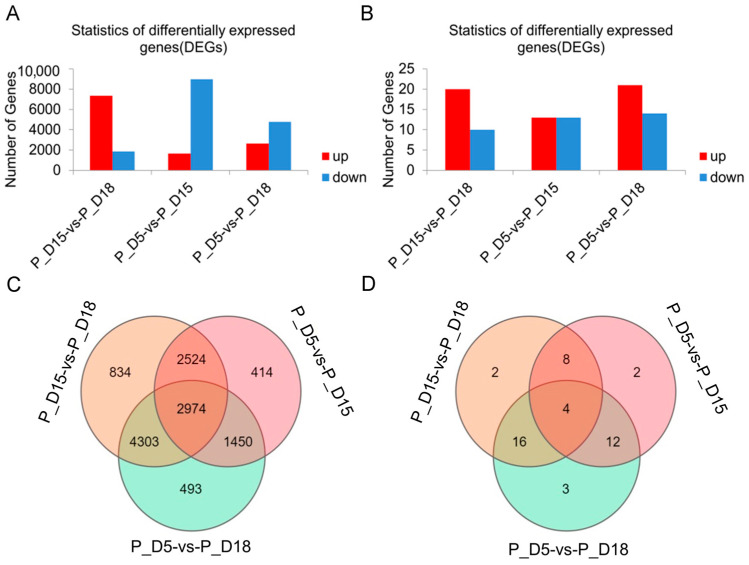
Differential abundance of mRNAs and miRNAs analyzed in exosomes isolated from the 5th day, 15th day, and 18th day of pregnancy. (**A**) mRNA. (**B**) miRNA. The Venn diagram shows the differential expression of mRNA (**C**) and miRNA (**D**).

**Figure 3 ijms-24-02799-f003:**
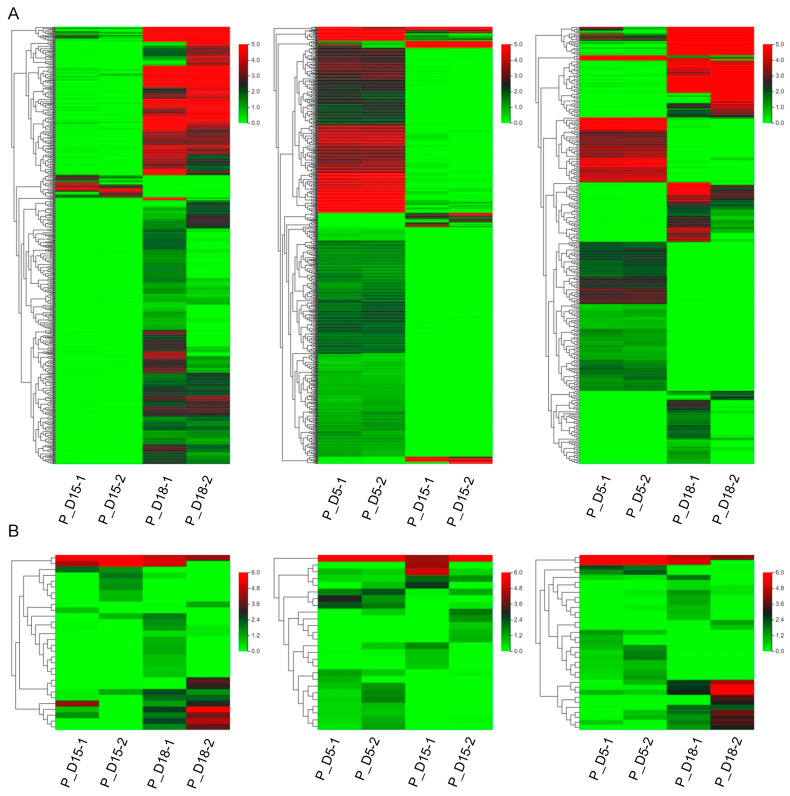
Heatmap of differential abundance of mRNAs (the mRNAs with a 14-fold difference in expression were selected for cluster analysis) and miRNAs. (**A**) mRNA. (**B**) miRNA. The order is P_D15 vs. P_D18, P_D5 vs. P_D15, P_D5 vs. P_D18.

**Figure 4 ijms-24-02799-f004:**
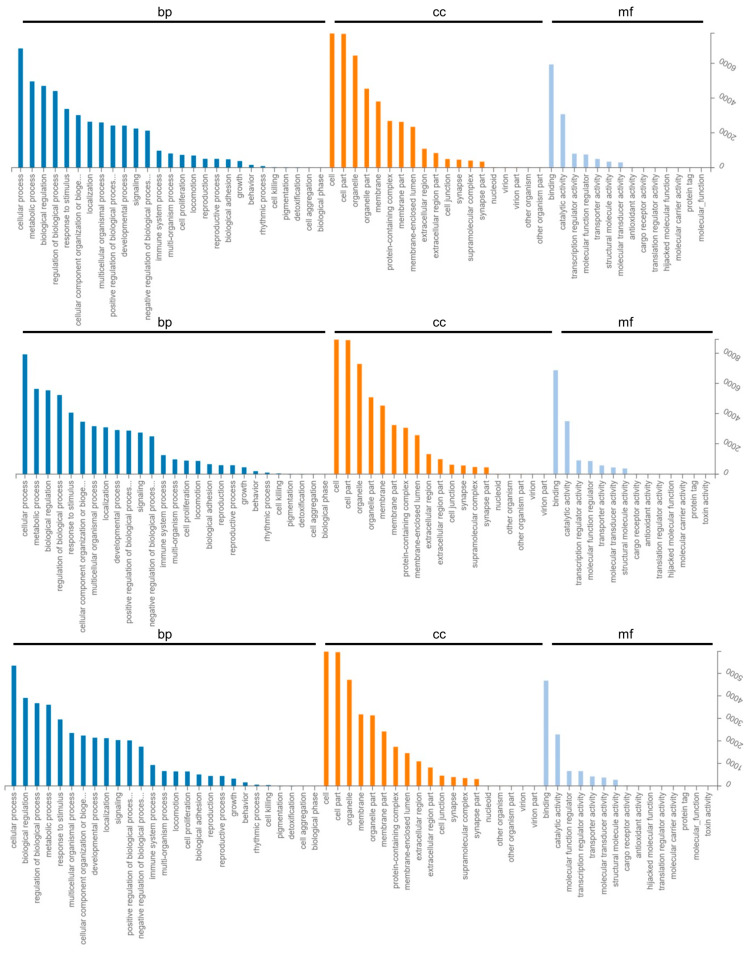
Gene Ontology classifications of differentially loaded mRNAs analyzed in exosomes. The order is P_D15 vs. P_D18, P_D5 vs. P_D15, P_D5 vs. P_D18. bp: Biological Process. cc: Cellular Component. mf: Molecular Function.

**Figure 5 ijms-24-02799-f005:**
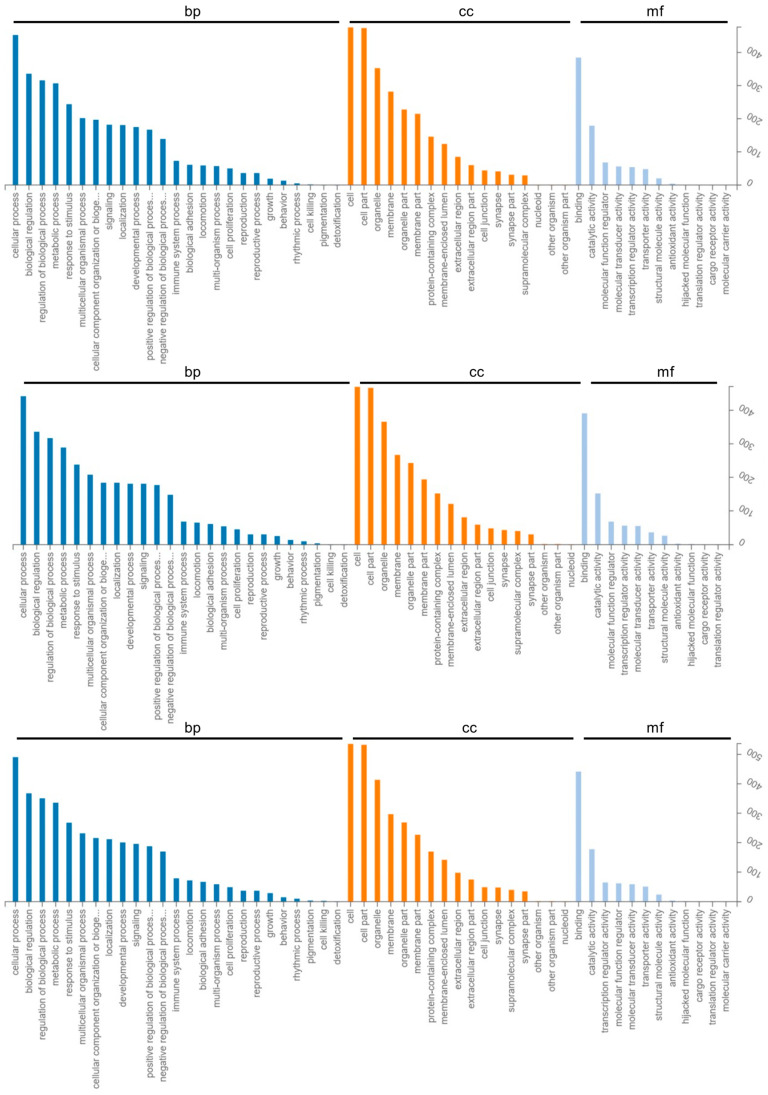
Gene Ontology classifications of target genes of differentially loaded miRNAs analyzed in exosomes. The order is P_D15 vs. P_D18, P_D5 vs. P_D15, P_D5 vs. P_D18. bp: Biological Process. cc: Cellular Component. mf: Molecular Function.

**Figure 6 ijms-24-02799-f006:**
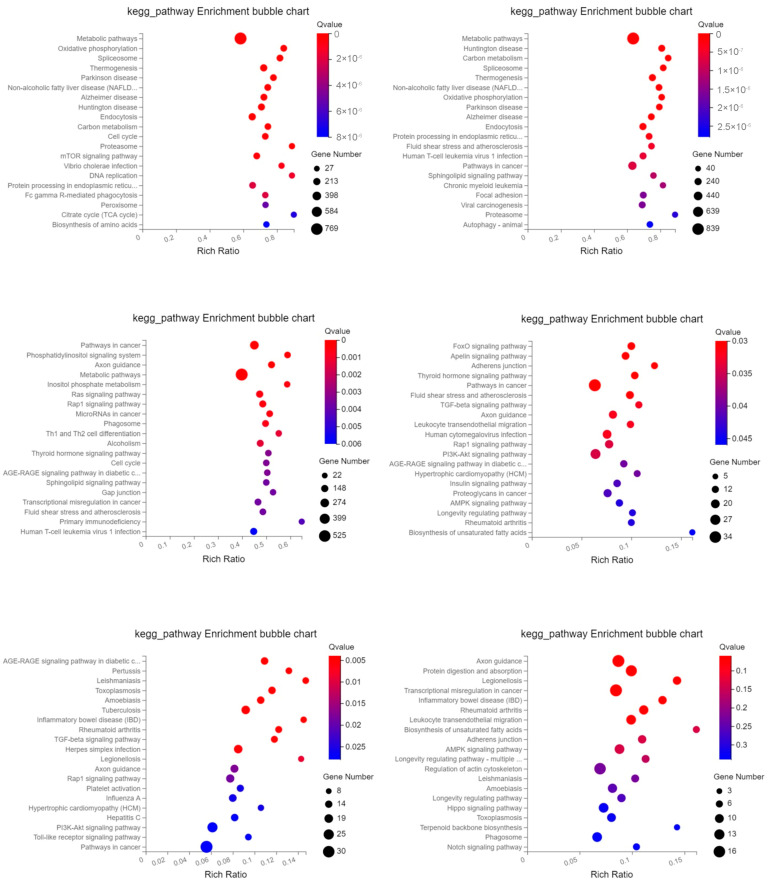
Top 20 KEGG pathways of differentially expressed mRNAs and miRNAs target genes. The *x*-axis indicates the number of unique sequences assigned to a specific pathway and the *y*-axis indicates the KEGG pathway. The size of the circle represents the number of genes enriched in the pathway. The order is P_D15 vs. P_D18, P_D5 vs. P_D15, P_D5 vs. P_D18 (mRNA) and P_D15 vs. P_D18, P_D5 vs. P_D15, P_D5 vs. P_D18 (miRNA).

**Figure 7 ijms-24-02799-f007:**
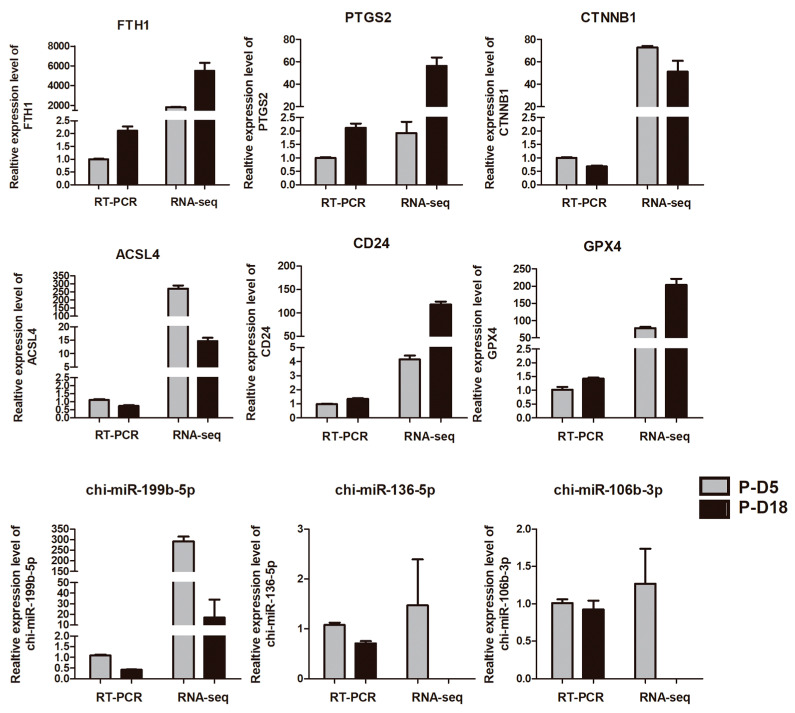
Verification of mRNA and miRNA expression levels. The relative expression level of randomly chosen dem in uterine cavity fluid exosomes from P_D5 and P_D18 was identified in the RT-PCR and RNA-seq. Data are represented as mean ± SD (*n* = 2).

**Figure 8 ijms-24-02799-f008:**
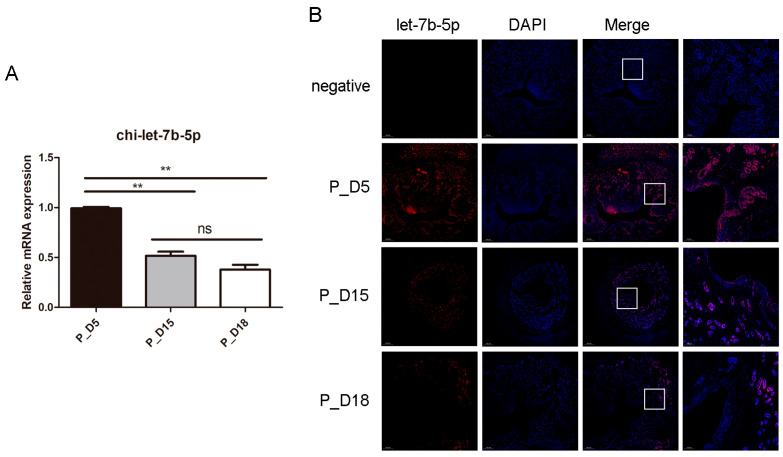
Expression and localization of chi-let-7b-5p in goat uterine tissue. (**A**) Expression of chi-let-7b-5p in goat uterine tissue. Data are represented with mean standard deviation from three independent experiments, ** *p* < 0.01. (**B**) In situ hybridization was used to detect chi-let-7b-5p in goat uterus. Scale bar = 500 µm and 100 µm.

**Figure 9 ijms-24-02799-f009:**
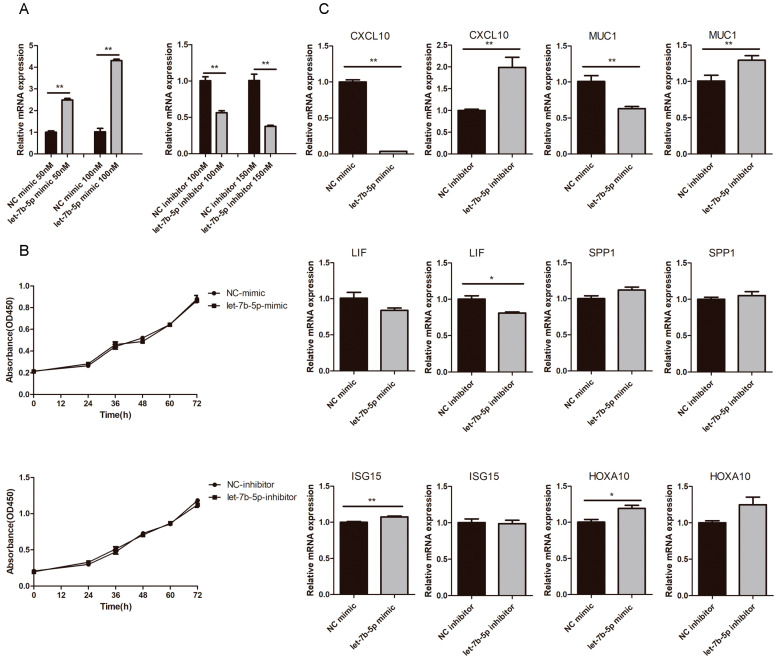
Effect of chi-let-7b-5p on gEEC after transfection with chi-let-7b-5p mimic and inhibitor. (**A**) Relative expression level of chi-let-7b-5p. (**B**) Effect of chi-let-7b-5p on the proliferation of gEEC. (**C**) The expression levels on *CXCL10*, *MUC1*, *LIF*, *SPP1*, *ISG15*, and *HOXA10*. Data are represented with mean standard deviation from three independent experiments, ** *p* < 0.01 and * *p* < 0.05.

**Figure 10 ijms-24-02799-f010:**
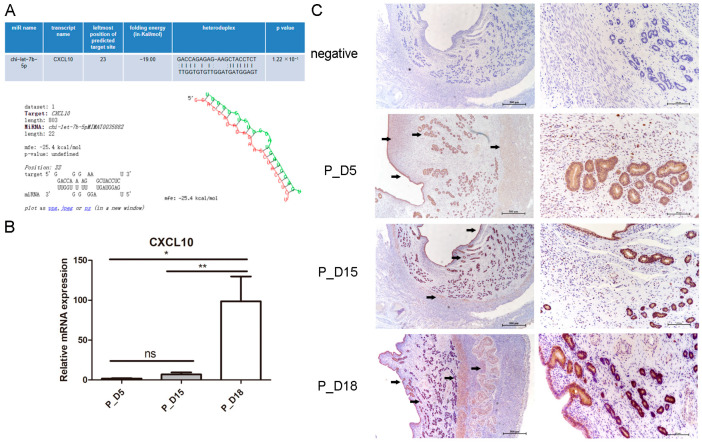
Expression and localization of CXCL10 in goat uterine tissue. (**A**) The predicted binding site of chi-let-7b-5p in the 3′UTR of CXCL10. (**B**) Expression of *CXCL10* in goat uterine tissue. Data are represented with mean standard deviation from three independent experiments, ** *p* < 0.01 and * *p* < 0.05. (**C**) Immunohistochemistry was used to detect CXCL10 in goat uterus. Scale bar = 500 µm and 100 µm.

**Figure 11 ijms-24-02799-f011:**
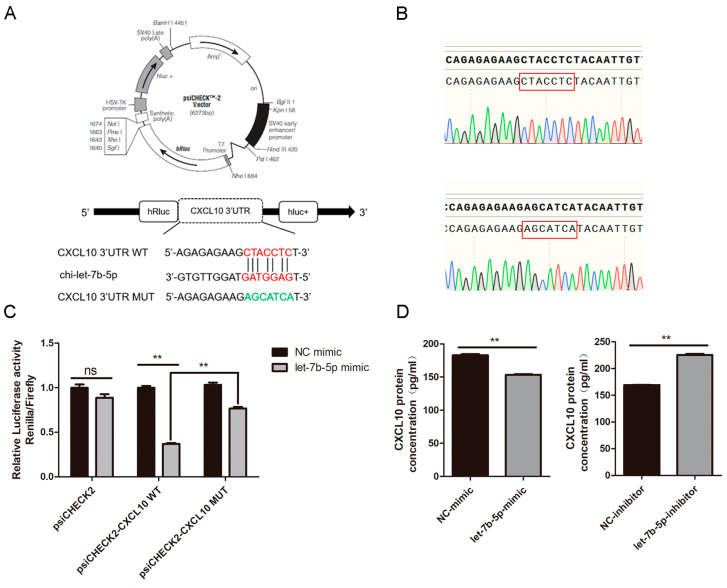
Chi-let-7b-5p targets the 3′UTR of CXCL10. (**A**) Design of double luciferase reporter plasmid and mutant plasmid. The CXCL10 3′ UTR sequence amplified by PCR contained the binding site with chi-let-7b-5p, and the CXCL10 3’ UTR MUT sequence had a mutation with the binding site with chi-let-7b-5p. (**B**) The Sanger sequencing result of sequences of wild-type and mutant vectors. (**C**) chi-let-7b-5p mimic or NC mimic was co-transfected with psiCHECK2, psiCHECK2-CXCL10 WT or psiCHECK2-CXCL10 MUT into 293T cells. Luciferase activity was measured after 36 h. Data are represented with mean standard deviation from three independent experiments. (**D**) gEEC was transfected with the chi-let-7b-5p mimic or inhibitor and the corresponding NC. The protein expression of CXCL10 was detected by ELISA. Data are represented with mean standard deviation from three independent experiments, ** *p* < 0.01.

**Table 1 ijms-24-02799-t001:** This is a table. Tables should be placed in the main text near to the first time they are cited.

Name	Forward (5′ → 3′)	Reverse (5′ → 3′)
*CXCL10*	CTGCCCACGTGTCGAGATTA	TGCCTCTTTCCGTGTTCGAG
*MUC1*	TACTCTGCCTACCACACCCA	CTGGACTCTCAGCAGACGTG
*LIF*	TGCTGCCTTACTTGGGTGAG	GCCTTTCAAGGGCCTCTCTT
*SPP1*	TGTTAAAGCAGGGTGGGAGAC	AGGGTGTTACCATGAAGCCAC
*ISG15*	GACACCAGAACCCACGG	GGAAAGCAGGCACATTGA
*HOXA10*	GTACCTTACTCGAGAGCGGC	TTGCCTGGAGCTTCATCAGG
*GAPDH*	TCTGCTGATGCCCCCATGTT	TGACCTTGCCCACGGCCTTG
chi-let-7b-5p	GCGCGTGAGGTAGTAGGTTGT	AGTGCAGGGTCCGAGGTATT
U6	CTCGCTTCGGCAGCACA	AACGCTTCACGAATTTGCGT
chi-let-7b-5p stem loop primer	GTCGTATCCAGTGCAGGGTCCGAGGTATTCGCACTGGATACGACAACCAC	
*FTH1*	CGTGATGACTGGGAGAACGG	TTGTGCAGTTCCAGTAGCGA
*PTGS2*	TGTATCCCGCCCTTCTGGTA	CCGGCTTCTACCATGGTCTC
*CTNNB1*	AATCAGCTGGCCTGGTTTGA	GCTTGGTTAGTGTGTCAGGC
*ACSL4*	AACTTCGGCAGTGGACTCAC	CCGCAATCATCCATTCAGCC
*CD24*	GCTGCTCTTACCTACGCAGA	CAGAGTACCATCGCTTGCCT
*GPX4*	TGTGGTTTACGGATCCTGGC	CCCTTGGGCTGGACTTTCAT
chi-miR-199b-5p	GCTCGACGCCCAGTGTTTAGACTAT	miRNA qPCR 3′primer
chi-miR-136-5p	ACTCCATTTGTTTTGATGATGG	miRNA qPCR 3′primer
chi-miR-106b-3p	CCGCACTGTGGGTACTTGCT	miRNA qPCR 3′primer

## Data Availability

Not applicable.

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
