# Peer review of "Uterine Flushing Fluid-Derived Let-7b Targets CXCL10 to Regulate Uterine Receptivity in Goats during Embryo Implantation"

_ijms, 2023, doi:10.3390/ijms24032799_

Round 1
Reviewer 1 Report
The authors present the paper without a line number, which makes it difficult to review. Even so, the research is interesting and I recommend its publication after minor changes.
Results
I recommend that in Figure 1 the identification (P_D5, P_D15 and P_D18 ) be added in the appropriate row of the figure. This would avoid repeating it in the figure caption text.
2.1. Exosome identification
Line 8. should be 1.81×1010 particles/mL; please correct.
Figure 2 A and B. The graphs are titled Statistics differentially... But the statistical difference that might exist in graph A (P_D5-vs-P_D15) is not shown.
Figure 4 and 5 describe in the caption of each figure what it means (bp, cc and mf).
2.4. Expression and localization of chi-let-7b-5p
Lines 1-3 is an interesting discourse, but I think that the results section should be devoted to explaining the results of your research. And these lines could be adapted to the introduction or in the discussion.
Discussion
I suggest restructuring the discussion as the first 24 lines are of information that should go in the introduction and/or materials and methods section.
Lines 27 and 34 the authors refer to Saadeldin et al. and Gurung S et al. as both active and passive citations. Please check this, throughout the document it should be: e.g. (LINE 27) Saadeldin et al. [29], research...
Material and methods
4.1. Sample collection and exosome isolation
Line 5. remove “ female or goat”. if we talk about gestation it is obvious that we refer to the female.
Author Response
International Journal of Molecular Sciences
18 January 2023
Dear Reviewers,
Please find enclosed the revised version of our manuscript entitled “Uterine flushing fluid-derived let-7b targets CXCL10 to regulate uterine receptivity in goats during embryo implantation” (ijms-2079180). The manuscript has been revised according to the comments of the reviewers. The revised manuscript resubmitted for your consideration.
We would like to thank the reviewers and editors of our manuscript for the interest they have expressed in our study and for their constructive comments. We have extensively revised the manuscript and highlighted changes in the revision to answer their criticisms and comments. Please find below a point by point response to the raised concerns. We hope that the revised manuscript is now suitable for publication in International Journal of Molecular Sciences.
Sincerely,
Dr. Dong Zhou
College of Veterinary Medicine,
Northwest A&F University, Yangling, China
Tel: 15809209367
- mail: zhoudong1949@nwafu.edu.cn
Response to Reviewer 1 Comments:
The authors present the paper without a line number, which makes it difficult to review. Even so, the research is interesting and I recommend its publication after minor changes.
Response: Thank you for your comments. We have added the line number to the manuscript.
Results
Point 1: I recommend that in Figure 1 the identification (P_D5, P_D15 and P_D18 ) be added in the appropriate row of the figure. This would avoid repeating it in the figure caption text.
Response 1: Thank you for providing us the valuable and constructive comments. We have added the identifiers (P_D5, P_D15, and P_D18) to the corresponding rows in Figure 1 and modified the figure caption text. (lines 107-109).
Point 2:
2.1. Exosome identification
Line 8. should be 1.81×1010 particles/mL; please correct.
Response 2: We are very sorry for our carelessness. We have modified it. (lines 100).
Point 3: Figure 2 A and B. The graphs are titled Statistics differentially... But the statistical difference that might exist in graph A (P_D5-vs-P_D15) is not shown.
Response 3: We are very sorry for our carelessness. We have modified it according to Reviewer’s comments and added the description of Venn Diagram. (lines 137-138 and 125-130).
Point 4: Figure 4 and 5 describe in the caption of each figure what it means (bp, cc and mf).
Response 4: We have added this part according to Reviewer’s comments. (lines 158-159 and 162-163).
Point 5:
2.4. Expression and localization of chi-let-7b-5p
Lines 1-3 is an interesting discourse, but I think that the results section should be devoted to explaining the results of your research. And these lines could be adapted to the introduction or in the discussion.
Response 5: Thank you for providing us the valuable and constructive comments. We have added it to the introduction (line 76-79) and rewritten the first sentence (line 197-199).
Discussion
Point 1: I suggest restructuring the discussion as the first 24 lines are of information that should go in the introduction and/or materials and methods section.
Response 1: We agree with the comments. We have added the first 24 lines to the introduction and M&M. The introduction and M&M sections have been carefully rewritten. (line 43-48 and 388-393).
Point 2: Lines 27 and 34 the authors refer to Saadeldin et al. and Gurung S et al. as both active and passive citations. Please check this, throughout the document it should be: e.g. (LINE 27) Saadeldin et al. [29], research...
Response 2: We have modified it according to reviewer’s comments. (lines 281 and 288).
Material and methods
Point 1:
4.1. Sample collection and exosome isolation
Line 5. remove “female or goat”. if we talk about gestation it is obvious that we refer to the female.
Response 1: We agree with the comment. It has been removed in the revised manuscript.

Reviewer 2 Report
The authors report that "immunohistochemical results showed that it was located primarily in endometrial epithelial cells, luminal epithelial cells, stromal cells, and deep muscle layers." What do the authors call "endometrial epithelial cells"? The luminal epithelium lines the uterus internally and is also considered endometrial epithelium. it is necessary to clarify. Immunohostochemistry photos need to be improved in their resolution. It is not possible to identify what is reported in the results.
Author Response
International Journal of Molecular Sciences
18 January 2023
Dear Reviewers,
Please find enclosed the revised version of our manuscript entitled “Uterine flushing fluid-derived let-7b targets CXCL10 to regulate uterine receptivity in goats during embryo implantation” (ijms-2079180). The manuscript has been revised according to the comments of the reviewers. The revised manuscript resubmitted for your consideration.
We would like to thank the reviewers and editors of our manuscript for the interest they have expressed in our study and for their constructive comments. We have extensively revised the manuscript and highlighted changes in the revision to answer their criticisms and comments. Please find below a point by point response to the raised concerns. We hope that the revised manuscript is now suitable for publication in International Journal of Molecular Sciences.
Sincerely,
Dr. Dong Zhou
College of Veterinary Medicine,
Northwest A&F University, Yangling, China
Tel: 15809209367
mail: zhoudong1949@nwafu.edu.cn
Response to Reviewer 2 Comments:
Point 1: The authors report that "immunohistochemical results showed that it was located primarily in endometrial epithelial cells, luminal epithelial cells, stromal cells, and deep muscle layers." What do the authors call "endometrial epithelial cells"? The luminal epithelium lines the uterus internally and is also considered endometrial epithelium. it is necessary to clarify. Immunohostochemistry photos need to be improved in their resolution. It is not possible to identify what is reported in the results.
Response 1: We are very sorry for our carelessness. Endometrial epithelial cells include luminal epithelial cells and glandular epithelial cells. Here, immunohistochemical staining results showed that CXCL10 was mainly localized in goat endometrial cavity epithelial cells, glandular epithelial cells, stromal cells, and deep muscle layer. We have modified it. Moreover, we have improved the resolution of the immunohistochemical image, which has been modified in Figure 10.

Reviewer 3 Report
The authors have assessed the miRNA and mRNA cargo of uterine flushings derived "exosomes" and then tried to explain how one of the miRNA could affect the endometrial cell functions specially the epithelial cells.
The MS is well written , however there are few major concerns and concerns would like to get clarifications.
-The authors have used ultracentrifugation /differential ultracentrifugation to enrich the exosomes, and then the characterization of exosomes has been performed using only the NTA and TEM. How certain all the particles enriched represent exosomes ?. Only by NTA with nano-fcm and imaging with TEM do not explain / support the presence of exosomes . Therefore , authors need to full fill ISEV guidelines on characterizing EVs/exosomes and need to give marker based evidences to prove the enrichment of exosomes.
- Whole transcriptomic and miRNA analysis using NGS have been performed using n=2 for each stage of the pregnancy. What were the criterion used in statistics in bioinformatics used in the analysis ?.
- Since the data are coming from only two samples per stage, authors need to valdiate the miRNA and mRNA expression data using qPCR. Specially the mRNA expression since it is now well accepted that exosomes do not carry full length functional mRNA ,rather they carry fragmented mRNA. Moreover, a miRNA extraction kit which is known to isolated RNA in small forms has been used in total RNA extraction which does not favour enriching long mRNA ligands as well. Thus, validation of omics using at least few targets from miRNA and mRNA expression in exosomes is required.
-The enriched exosomes may contain EVS from both embryo and endometrium origin. However, the authors only have tested the impact of Let-7 miRNA on endometrium only. At least in the discussion , an appreciation has to be done from the contribution from embryo/conceptus.
Author Response
International Journal of Molecular Sciences
18 January 2023
Dear Reviewers,
Please find enclosed the revised version of our manuscript entitled “Uterine flushing fluid-derived let-7b targets CXCL10 to regulate uterine receptivity in goats during embryo implantation” (ijms-2079180). The manuscript has been revised according to the comments of the reviewers. The revised manuscript resubmitted for your consideration.
We would like to thank the reviewers and editors of our manuscript for the interest they have expressed in our study and for their constructive comments. We have extensively revised the manuscript and highlighted changes in the revision to answer their criticisms and comments. Please find below a point by point response to the raised concerns. We hope that the revised manuscript is now suitable for publication in International Journal of Molecular Sciences.
Sincerely,
Dr. Dong Zhou
College of Veterinary Medicine,
Northwest A&F University, Yangling, China
Tel: 15809209367
mail: zhoudong1949@nwafu.edu.cn
Response to Reviewer 3 Comments:
The authors have assessed the miRNA and mRNA cargo of uterine flushings derived "exosomes" and then tried to explain how one of the miRNA could affect the endometrial cell functions specially the epithelial cells.
The MS is well written , however there are few major concerns and concerns would like to get clarifications.
Point 1: The authors have used ultracentrifugation /differential ultracentrifugation to enrich the exosomes, and then the characterization of exosomes has been performed using only the NTA and TEM. How certain all the particles enriched represent exosomes ?. Only by NTA with nano-fcm and imaging with TEM do not explain / support the presence of exosomes. Therefore , authors need to full fill ISEV guidelines on characterizing EVs/exosomes and need to give marker based evidences to prove the enrichment of exosomes.
Response 1: Thank you for providing us the valuable and constructive comments. We have supplemented the Western Blot analysis of exosome according specific marker proteins in Figure 1 and modified the figure caption text. More detailed the M&M and results section has been described (lines 102-103, 107-109 and 426-435) .
Point 2: Whole transcriptomic and miRNA analysis using NGS have been performed using n=2 for each stage of the pregnancy. What were the criterion used in statistics in bioinformatics used in the analysis ?
Response 2: We acknowledge the reviewer’s concern. We employed two methods that correct P-values to Q-values for the sequencing data developed by Benjamini, Hochberg, Storey, and Tibshirani. Significantly differentially expressed genes (SDEGs) were classified and screened as genes with a Q-value≤0.001 and a difference greater than two times the control. (lines 567-571).
Point 3: Since the data are coming from only two samples per stage, authors need to valdiate the miRNA and mRNA expression data using qPCR. Specially the mRNA expression since it is now well accepted that exosomes do not carry full length functional mRNA, rather they carry fragmented mRNA. Moreover, a miRNA extraction kit which is known to isolated RNA in small forms has been used in total RNA extraction which does not favour enriching long mRNA ligands as well. Thus, validation of omics using at least few targets from miRNA and mRNA expression in exosomes is required.
Response 3: Thank you for providing us the valuable and constructive comments. We supplemented the miRNA and mRNA expression data using qPCR. More statement describing the qPCR analysis and the results were added in the revised manuscript (lines 190-194).
Point 4: The enriched exosomes may contain EVS from both embryo and endometrium origin. However, the authors only have tested the impact of Let-7 miRNA on endometrium only. At least in the discussion , an appreciation has to be done from the contribution from embryo/conceptus.
Response 4: We acknowledge the reviewer’s concern. We have carefully rewritten the discussion. “Additionally, some research has demonstrated that exosomes produced from embryos enhance the quantity and quality of blastocysts, while exosomes secreted by embryonic stem cells isolated from the inner cell mass encourage trophoblast cell migration [44] ADDIN EN.CITE. To ascertain let-7b-5p's role in the implantation of goat embryos, additional research is required(Line 362-366)”.

Round 2
Reviewer 3 Report
- The authors have satisfactorily addressed the querries raised.
- To improve the methodology description to make them clear, thus, authors can include the replicates/sample numbers used in their experiments e.g. how many n for each RT-PCR validation experiments, whether the samples were from same animals used for NGS or different animal cohort? At least include such information clearly in figure legends.
- still the authors can highlight the possibility of let 7 origin from embryo to enhance the endometrial micro-environment, specifically during the preimplantation period since many report that embryo can also prodcue let 7 miRNA (https://www.ncbi.nlm.nih.gov/pmc/articles/PMC2688687/)
Author Response
Response to Reviewer 3 Comments:
The authors have satisfactorily addressed the querries raised.
Point 1: To improve the methodology description to make them clear, thus, authors can include the replicates/sample numbers used in their experiments e.g. how many n for each RT-PCR validation experiments, whether the samples were from same animals used for NGS or different animal cohort? At least include such information clearly in figure legends.
Response 1: Thank you for providing us the valuable and constructive comments. We have supplemented information of the replicates in M&M section and figure legends according to the reviewer's comments. (Line 196-198; Line 218-221; Line 238-242; line 253-257; Line 280-283; Line379-380; Line513-514) .
Point 2: still the authors can highlight the possibility of let 7 origin from embryo to enhance the endometrial micro-environment, specifically during the preimplantation period since many report that embryo can also prodcue let 7 miRNA (https://www.ncbi.nlm.nih.gov/pmc/articles/PMC2688687/)
Response 2: Thank you for providing us the valuable and constructive comments. We have supplemented relevant content to the discussion section according to the reviewer's comments. “Numerous studies have demonstrated that let-7 miRNA [42, 43] can be produced by animal embryos at the pre-implantation stage and that the lin28b/let-7 axis can regulate the differentiation of fetal Treg cells [44]. Additionally, it has been demonstrated that the immune microenvironment during pregnancy is essential to the embryonic implantation process [45]. Let-7b-5p may come from embryos, and our research has demonstrated the direct targeted link between it and CXCL10. It also plays a critical function in embryo implantation by controlling the uterine immune milieu. More investigation is required to establish let-7b-5p's function in the implantation of goat embryos.” (Line 368-376).
